# Reinforcing Efficiency of Micro and Macro Continuous Polypropylene Fibers in Cementitious Composites

**Barzin Mobasher [1,*], Vikram Dey [2], Jacob Bauchmoyer [3], Himai Mehere [1] and Steve Schaef [4]**

[1] School of Sustainable Engineering and the Built Environment, Arizona State University, Tempe, AZ 85287, USA; hmehere@asu.edu

[2] Structural Designer, PK Associates Structural Engineers, Scottsdale, AZ 85250, USA; vikram.dey@asu.edu

[3] Structural Engineer, CDM Smith, Phoenix, AZ 85028, USA; jbauchmo@asu.edu

[4] Materials Engineer, Development Admixture Systems, Beachwood, OH 44133, USA; steve.schaef@basf.com

**\*** Correspondence: barzin@asu.edu; Tel.: +1-480-965-0141; Fax: +1-480-965-0557

**Abstract:** The effect of the microstructure of hydrophilic polypropylene (PP) fibers in the distribution of cracking associated with the strengthening and toughening mechanism of cement-based composites under tensile loading was studied. Using a filament winding system, continuous cement-based PP fiber composites were manufactured. The automated manufacturing system allows alignment of the fiber yarns in the longitudinal direction at various fiber contents. Composites with surface-modified hydrophilic macro-synthetic continuous polypropylene fibers and monofilament yarns with different diameters and surface structures were used. Samples were characterized using the tensile first cracking strength, post-crack stiffness, ultimate strength, and strain capacity. A range of volume fractions of 1–4% by volume of fibers was used, resulting in tensile first cracking strength in the range of 1–7 MPa, an ultimate strength of up to 22 MPa, and a strain capacity of 6%. The reinforcing efficiency based on crack spacing and width was documented as a function of the applied strain using digital image correlation (DIC). Quantitative analysis of crack width and spacing showed the sequential formation and gradual intermittent opening of several active and passive cracks as the key parameters in the toughening mechanism. Results are correlated with the tensile response and stiffness degradation. The mechanical properties, as well as crack spacing and composite stiffness, were significantly affected by the microstructure and dosage of continuous fibers.

**Keywords:** fiber-reinforced concrete; crack spacing; fiber; micro-fiber; tensile strength; toughness

## 1. Introduction

Development of strain-hardening cementitious composites (SHCC) using polypropylene (PP) fibers is a major breakthrough for a variety of applications in civil infrastructure systems. SHCC materials, such as textile reinforced concrete (TRC), exhibit high tensile strength, enhanced strain capacity, and ductility [1–3]. The superior mechanical properties offered by the polymeric based continuous fiber or textile system can be utilized as structural panels subjected to dynamic loads, such as impact and high speed, along with applications requiring blast resistance and fracture tolerance. SHCC systems could also be used as skin reinforcement laminates for the strengthening of unreinforced masonry walls, retrofit of existing structures, and beam–column connections [4–6]. The tensile hardening behavior is attributed to the fiber bridging effect, which stabilizes crack growth and opening at the expense of the formation of multiple, parallel fine cracks. This cracking network gives rise to high energy absorption, both under quasi-static and dynamic loading conditions. The post-crack stiffness

and the corresponding damage distribution may form with a variety of fiber systems and is governed by the fiber's ability to provide a sufficient degree of bond strength [7].

A class of SHCC materials made with polypropylene fibers with a high tensile ductility and stiffness retention over a large strain range is investigated in this study. Ductility enhancement is attractive from a cost point of view since polymeric fibers have a lower cost than steel, carbon, or other high-performance fibers; however, the efficiency of PP-based fibers is created in the form of developing composites with improved bond characteristics. Results are characterized by the improved bond characteristics of long and multifilament fibers, surface modifications, reduced diameter, and increased surface area of yarns. It is shown that proper mix proportioning results in excellent matrix properties [8–12].

Continuous unidirectional yarns were evaluated for two types of fiber compositions in this study. Effectiveness of the fiber–matrix bond interface in load transfer and distributed cracking in mechanical performance is addressed. Limitations in interfacial bond and low adhesion strength are major inefficiencies limiting the structural application of polymeric fibers in concrete materials. A combination of low organic–inorganic bond stiffness and strength limits the effectiveness of fiber–matrix stress transfer. Strength and toughness increases due to the increased aspect ratio in continuous fiber composites can be utilized in a variety of structural elements subjected to extreme loading conditions as discussed earlier [13].

Hydrophilic polymeric surfaces improve fiber performance and efficiency by affecting the bond stiffness and strength. Anchorage and bonding are also enhanced by geometrical modifications of the surface texture of the fiber [14]. Increasing the contact surface area by using small diameter filaments bundled into the form of yarn leads to additional bonding. In both these cases, the efficiency of the fiber performance is measured in the context of the fibers bridging over the cracks in the cementitious matrix, which subsequently de-bond and pullout, thus hindering the extension of cracks [15]. The fiber bridging and pullout force transmission reduces the crack tip stresses and increases toughness through energy dissipation [16]. The stress transmission through the bridging fibers is a major source of toughening and permits the initiation of new cracks, thus improving energy dissipation capacity of the composite [17].

Fiber length and orientation plays an important role in the mechanical response of cementitious composites. In order to eliminate the reduction factors due to length and orientation, unidirectional continuous composites were manufactured using a filament winding technique and tested in uniaxial tension. In the current study, two different polypropylene fiber types, namely macro-monofilaments and micro-multifilament yarns, at different dosages, are compared in terms of composite performance based on the tensile strength, crack spacing, and stiffness reduction as a function of measured strain. Matrix formulations consisted of blended cementitious matrices containing various proportions of Type II Ordinary Portland Cement (OPC), sand, and fly ash as a control matrix mix. Mechanical tests were performed under uniaxial tension, and three-dimensional digital image correlation (DIC) method and image analysis were used to quantify the damage mechanism and the non-uniform strain distributions. The distributed cracking mechanism was quantified by measuring the crack width and spacing and was further compared to the experimental stress–strain measures.

## 2. Experimental Program

Proprietary polypropylene yarns manufactured by BASF Construction Chemicals, Beachwood OH, USA were studied. A macrofiber labeled as MAC 2200CB (abbreviated as MAC in this study) is a commercially available monofilament macro-synthetic polypropylene fiber with an average diameter of 0.82 mm and pinched surface to improve the bond (see Figure 1a). It is used in cast-in-place concrete applications, such as slab-on-grade, pavements, bridge decks, and in precast concrete, mainly as a secondary reinforcement to restrain temperature cracking [18]. The second fiber evaluated in this study is a recently developed multifilament microfiber yarn with 500 thin filaments 40 microns in diameter and identified as MF 40 microfibers, as shown with two different magnifications in Figure 1b,c.

The effective yarn diameters of MF/MAC measured from the SEM images represent a surface to volume ratio of about 20.

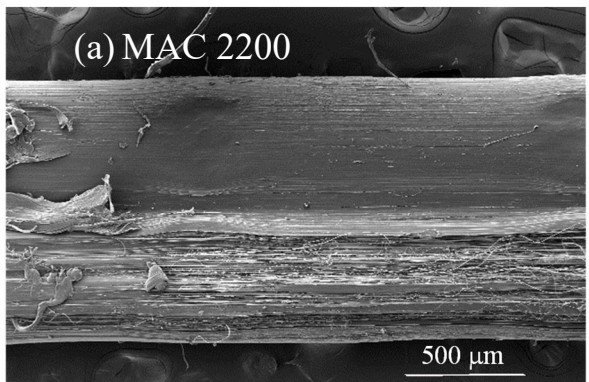
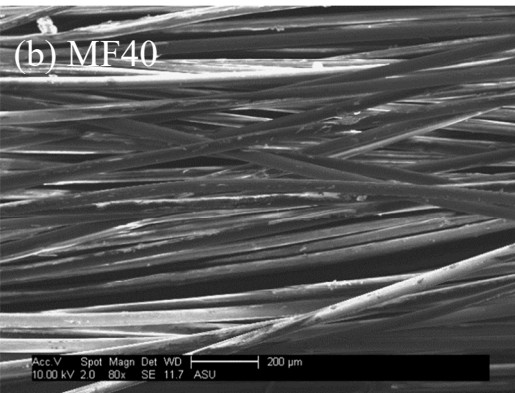

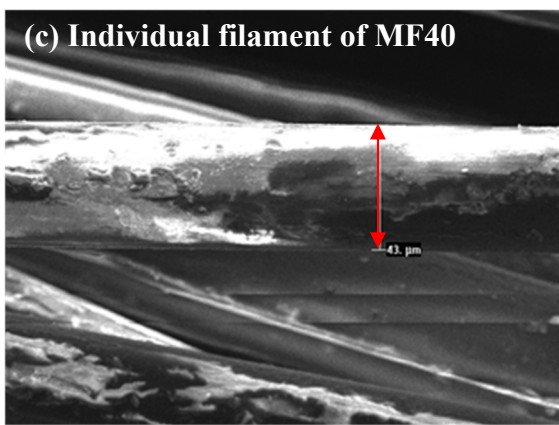

**Figure 1.** (**a**) Macro-synthetic MAC fiber, (**b**) multifilament fibrillated microfiber, and (**c**) diameter of an individual MF40 fiber.

Unidirectional composites were produced using the filament winding method shown in Figure 2a,b. Composites with continuous fibers allowed for measurement of reinforcement potential that is independent of the fiber length, delamination, or orientation effects. The experimental plan for mechanical tests is presented in Table 1 and includes tension tests on individual yarns and composite uniaxial tension. Testing variables included the fiber structure and content to study their affect on the tensile stress–strain response and damage parameters such as crack spacing and crack width [14,19].

### 2.1. Sample Preparation Using Filament Winding

A filament winding system was configured to fabricate continuous cement fiber laminates with aligned fiber yarns [13,17]. A computer-controlled system used stepper motors to pull the yarns and wind the sample on a mandrel. System components included the feed section, guidance assembly, and take-up mandrel. Labor-intensive tasks in production and panel making were reduced through this automated system. Servo-drives were programmed for automation of three sections of fiber feed, guide (fiber impregnation), and the take-up (molding) sections, as shown in Figure 2a.

The various sections included the stepper motors, positioning encoders, limit switches for safe interlocking, and a computer with control software interface, as shown in Figure 2b. The feed section used a spool of fibers that would unwind and was immersed into a wetting tank prior to immersion in an impregnation chamber. The sample was then wound on a rotating mold. Using a LabView© (2014, National Instruments, Austin, TX, USA) interface, a closed-loop system controls two stepper motors to

feed and slide the yarns through and rotate the mandrel. The stepper motors in the take-up section controlled the winding, pulling, and transverse sliding of the composites.

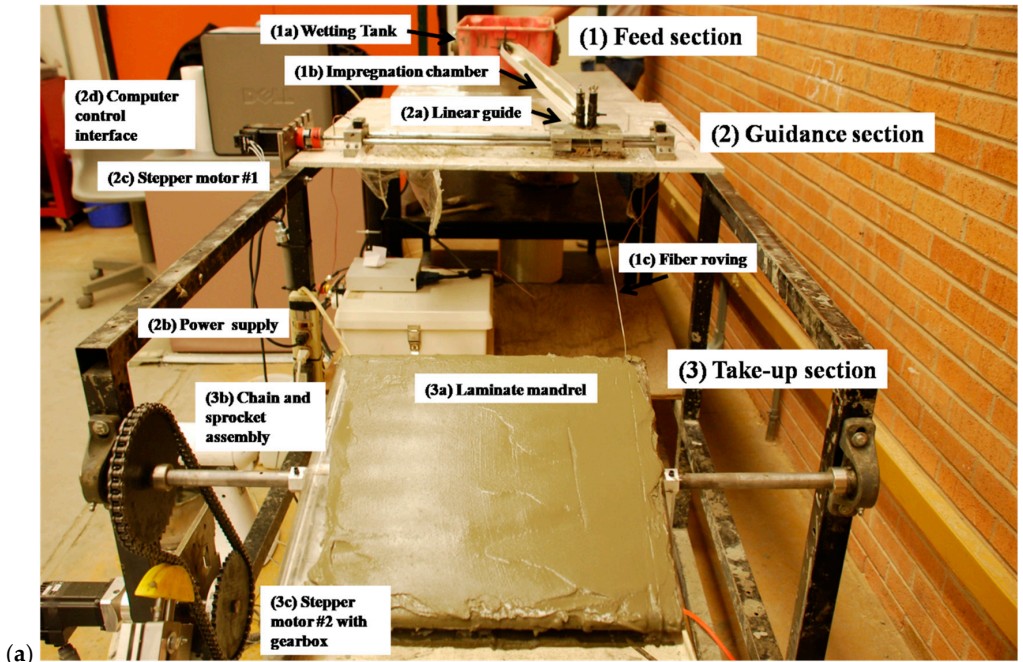

(a)

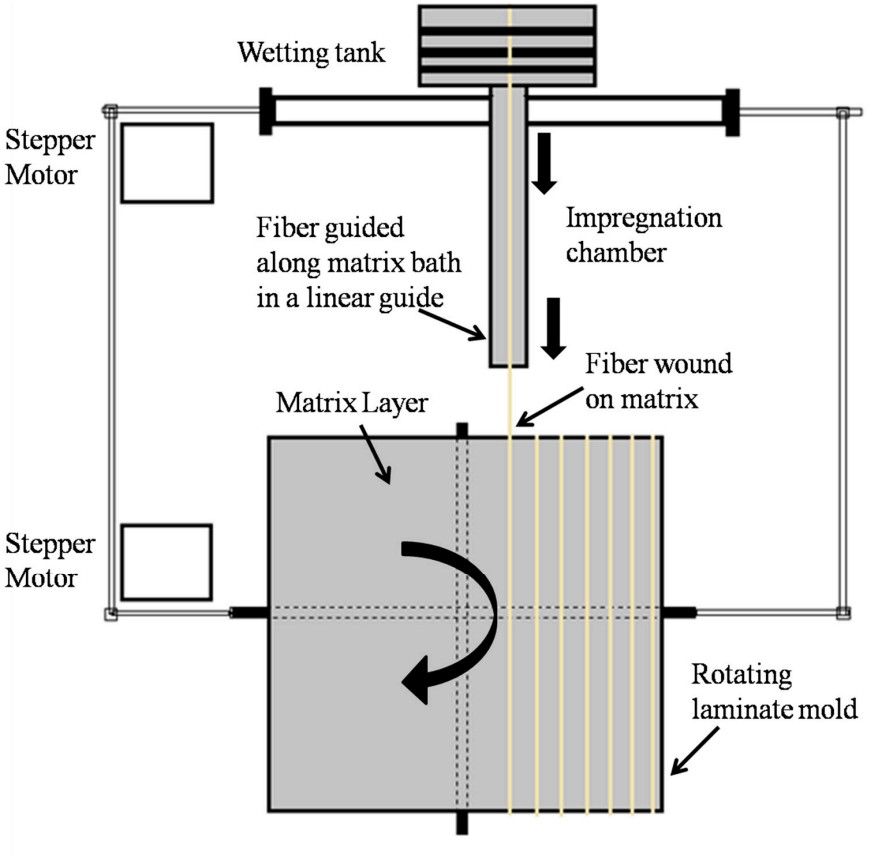

(b)

**Figure 2.** (**a**) Filament winding setup with the impregnation chamber, fiber guide section, and rotating mandrel; and (**b**) schematics of the steps section.

## 2.2. Mix Design

The control mix consisted of Portland cement, fly ash, and fine silica sand was used as the basic formulation of the composite design listed in Table 1. The control mortar mix design used a blend of 48% Portland cement type I/II, and 7% by weight of class F fly ash, a sand to cementitious solid ratio of 45% by weight, and water to binder ratio of 0.35. A naphthalene base high range water reducer manufactured by BASF was used at a dosage of 0.03% by weight of cement. Samples were made with the two polymeric fibers introduced earlier, namely macrofibre MAC and multifilament MF fibers, using 1%, 2.5%, and 4% volume fractions. Direct tension tests were conducted on a minimum of four replicate samples for each mix design.

**Table 1.** Summary of Tension specimens with continuous fibers.

| Test Type | Yarn Type | Sample Variables | Curing | Yarn $V_f$% |
|---|---|---|---|---|
| Fiber Tension | MAC | 150, 200, and 250 mm | N/A | |
| Fiber Tension | MF40 | 150, 200, and 250 mm | N/A | |
| Composite Tension | MAC | Volume fraction | 28 days | 1.0, 2.5, 4.0 |
| Composite Tension | MF40 | Volume fraction | 7, 28 days | 1.0, 2.5, 4.0 |

## 3. Testing Program

### 3.1. Tensile Response of Fibers

Fiber tension tests were conducted under displacement control mode to measure elastic modulus, strain capacity, ultimate strength, toughness, and mode of failure. The setup is shown in Figure 3a with a specimen under the applied load. An actuator displacement rate of 0.4 mm/min was used. Preliminary tests were conducted using fiber lengths of 150, 200, and 250 mm to address the length effect. Follow up studies used a sample length of 150 mm and a minimum of five replicate samples per series. The load was measured using a load cell rated at a capacity of 1300 N, while the elongation was recorded by an extensometer with a 50 mm gage length. A close-up view of the failed specimens with the extensometer attached is shown in Figure 3b,c.

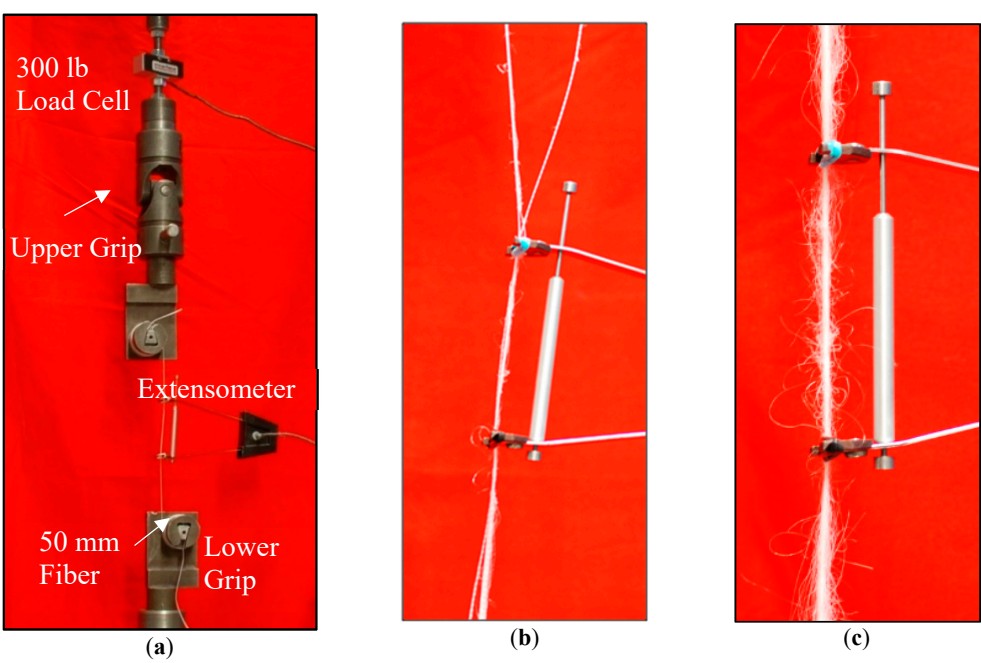

**Figure 3.** (**a**) Test setup for yarn tension tests, (**b**) failed MAC fiber, and (**c**) failed MF40 fiber.

Stress–strain behavior for the two types of MAC and MF fiber yarns are presented in Figure 4 and a summary of test results is given in Table 2. The initial stress–strain curve started with a stiff response up to a stress level of about 5–7 MPa.Beyond that level, the stiffness decreased due to fiber yielding. The general behavior was linear for the monofilament samples up to the failure; however, significant nonlinearity was observed for the microfiber yarn. Gradual transition of the stress–strain response of microfiber yarns to a nonlinear behavior started from 50% ultimate strain capacity without a clearly marked yield point. Beyond this level, the stiffness reduced gradually until failure.

Table 2 summarizes the single fiber tensile test results for both fiber types representing values of initial elastic modulus, $E_1$, and a post yield modulus, $E_2$. The macro-synthetic fiber, MAC, hadcomparatively higher initial and post-yield modulus, and showed a sudden failure compared to a progressive failure of the individual filaments of MF40 yarn. The ultimate strength was reached in a gradual manner for MF40 as opposed to a sharp end for MAC. Figure 3b,c show the failed MAC and MF40 specimens, respectively. With a strain capacity in the range of 12%, microfibers deformation was almost twice as much as the monofilament fibers, as shown in Figure 4. Compared to MAC, the MF40 microfiber exhibited significant crazing. This response was more pronounced when the strain wasmeasured using the actuator signal, as shown in Figure 4b, which also included the relative slipping of the individual filaments past each another, resulting in an apparent tensile strain as high as 75% for the MF series. These slip mechanisms lead to a higher strain capacity of the MF compared to MAC fibers.

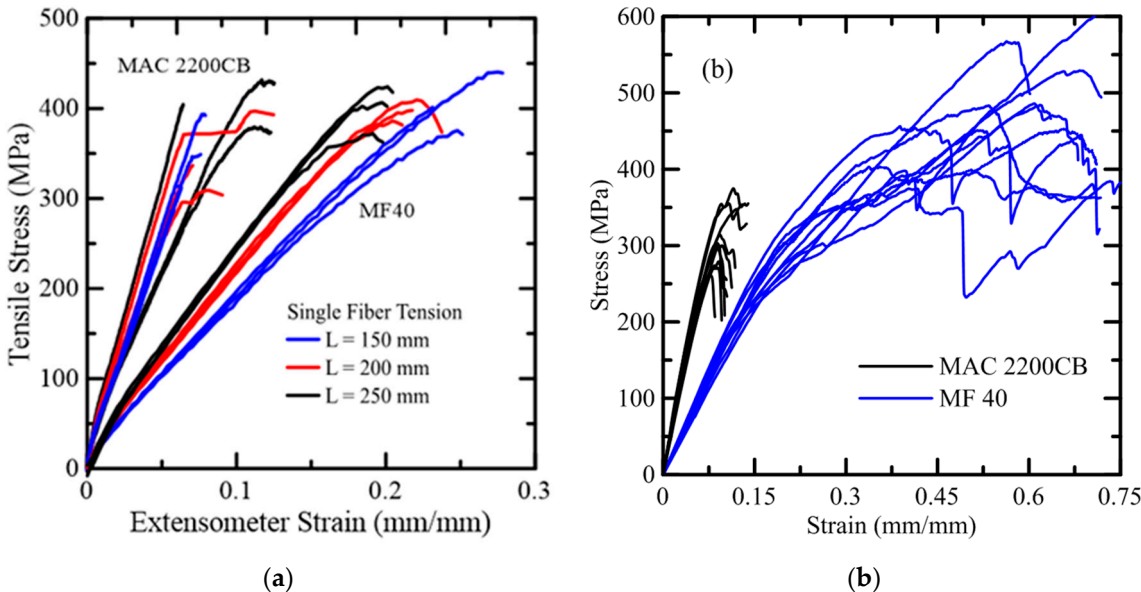

**(a)** **(b)**

**Figure 4.** (**a**) Effect of sample length on the initial response of the MAC and MF40 fibers. (**b**) Tensile stress versus actuator strain comparing MAC and MF40 failure for fiber length of 150 mm.

**Table 2.** Single fiber tests result for MAC and MF40 fibers, gauge length 150 mm.

| Fiber | | Max Load | Max Elongation | Tensile Strength | Elastic Modulus, $E_1$ | Post-Yield Modulus, $E_2$ | Work to Fracture |
|---|---|---|---|---|---|---|---|
| | | N | mm | MPa | MPa | MPa | J |
| MAC | Avg. | 245.3 | 4.4 | 394 | 9239 | 4566 | 0.70 |
| | Std Dev. | 20.0 | 0.9 | 32.8 | 1813 | 918.3 | 0.25 |
| MF | Average | 293.8 | 6.3 | 405 | 4985 | 3058 | 1.59 |
| | Std Dev. | 33.0 | 0.7 | 45.5 | 1112 | 479.3 | 0.21 |

The difference in strain capacity between the two fiber compositions resulted in the toughness of MF being twice that of MAC and is attributed to the structure of multifilament yarns, which by

distributing the damage among multiple fibers, promoted a progressive failure mechanism. This led to a 43% higher strain capacity than the macro MAC fiber, which was an inherently stiffer system (MAC = 9.2 GPa, MF40 = 5 GPa), as shown in Table 2. The post-yield reduced modulus for MAC was 4.6 GPa, which was 50% higher than post-yield modulus of MF40, which was at 3 GPa. Due to their strain capacity, finer MF40 fibrils required as much as 220% higher work to fracture. Multi-filament yarns uniformly distributed the load within the filaments, which failed sequentially over the failure strain range.

### 3.2. Tension Tests on Continuous Fiber Composites

A closed-loop servo-hydraulic test system, as shown in Figure 5, was used in actuator displacement control mode to conduct direct tension tests on the SHCC composites. Test coupons had nominal dimensions of 300 × 62 × 13 mm. The specimen was held using hydraulic grips with the pressure maintained between 1.7 and 2 MPa. Elongation was measured along a gage length of 90 mm using two linear variable differential transformers (LVDT) of 6 mm range and their average response was recorded along with the applied load and actuator displacement.

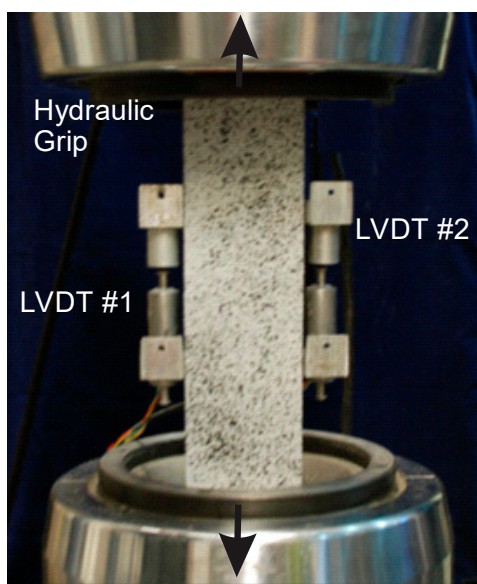

**Figure 5.** Tensile testing setup used to measure the characteristic response shown on the typical stress–strain response.

The characteristic stress–strain, crack spacing, and crack widening responses are summarized in Figure 6a–c. The observed stages of damage zones have been identified schematically in this figure and used in the discussion of results. The typical stress–strain response is predominantly linear up to point A, which is represented as the bend over point (BOP), this is referred to as Stage I. This is followed by the formation of the first crack in the specimen and initiation of Stage II. Between points B and C, there isthe formation of multiple distributed cracks and the initiation of fiber–matrix debonding. The bond exhibited by fibers resulted in crack bridging as the key toughening mechanism, which prevented the localization of individual cracks and promoted additional cracking. When a sufficient number of cracks had formed, stage III was initiated wherein crack saturation and widening of existing cracks occurred, leading to localized damage between points C and D. Crack saturation occurred due to limitations of the bond when stress in the matrix was insufficient to cause further cracks. Finally, in stage IV, tensile failure, fiber debonding, and slip occurred, and they were irreversible [7]. Beyond point D, the specimen significantly lost its load-carrying capacity and ultimately underwent complete failure.The experiments addressed the composite performance of the laminates using the correlation between the fiber–matrix bond, multiple cracking, crack widening, and crack saturation density.

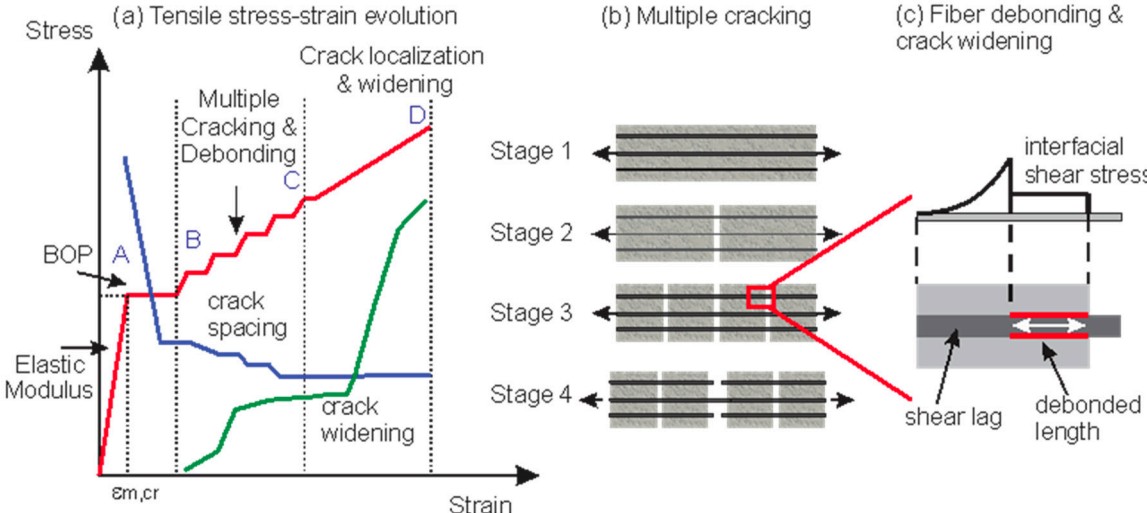

**Figure 6.** (**a**) Tensile testing results and regions of characteristic response shown on the typical stress–strain response, (**b**) four stages of linear elastic, cracking, multiple cracking, localization, and pullout, and (**c**) the interface debonding and pullout which contribute to crack widening.

Figure 7 shows the tensile response of the MF 40 fiber composite at two different fiber contents of 1% and 2.5% for curing durations of 7 and 28 days. The stress–strain response can be classified using the four stages as defined in Figure 6. In stage I, due to the linear behavior of matrix and fiber layers, the average strain in the longitudinal direction was uniform for the composite, fiber, and matrix. An increasing load initiated matrix cracking and stress was transferred to the fiber. Depending on the fiber content and bond, the first cracking was initiated in the form of a micro-crack and propagated along the width of the specimen at the bend over point (BOP) stress level, which is associated with the tensile strength of the matrix. Figure 7 indicates that BOP was directly correlated to the fiber content and curing age, and characterized by the elastic modulus, first crack strength, and strain.

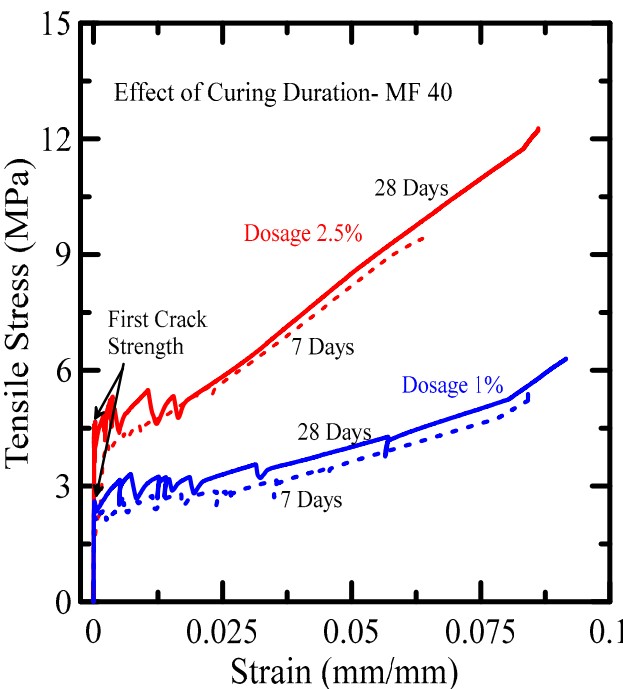

**Figure 7.** Effect of curing duration on the tensile response of microfilament based composites.

After initiation, a micro-crack may propagate in a stable manner due to fiber bridging, leading to a gradual reduction of matrix stiffness. The overall aspect of fiber contents is addressed in Figure 8a,b. The tensile stress–strain response of MAC and MF40 composites with a control matrix at various fiber dosages are presented at distinct stages of cracking for replicate test coupons. Higher fiber content increased the first crack strength [15] since a higher energy demand for the matrix crack propagation is imposed, which increased the apparent first crack strength. Formation of multiple parallel cracks was designated as stage II, as shown in Figure 6a. The dominant strain hardening behavior initiated after the first crack with additional parallel cracks occurring sequentially since the stiffness of the fiber phase allows for it to carry the load released by the matrix failure. The parallel cracks formed until the minimum crack spacing was reached, which correlated with the overall stiffness of the fibers, as shown in Figure 8c,d. The crack width and spacing were affected by the bond parameters and fiber content. This stage ended with fiber debonding as new crack formation seized and the existing cracks widened, matrix stress reduced, and fibers began to either get pulled out from the matrix or underwent fracture. In samples with 2.5% fiber, a 20% increase in BOP stress, and 35% increase in ultimate strain, which led to a 30% increase in UTS and 75% increase in toughness from 7 to 28 days of curing, was observed.

Summary results are presented in Table 3. The effect of fiber content on the first crack strength was more pronounced for MF40 composites in comparison to the MAC fibers. This was because of the bond surface area and distribution of thefibers throughout the matrix reducing the fiber to fiber specific spacing and reducing the minimum flaw size. The fiber bridging effect on the growing cracks was enhanced due to their distribution. The switch over from Stage I to II depended on the dosage of fibers available for bridging. The first cracking stress was higher for laminates with a higher fiber volume fraction for both MAC and MF40. At the same time for each category, the increasing volume fraction increasedthe first crack strength. MAC fibers showed an average 1.4 MPa stress level for 1% and 2.5% fiber dosages, while at 4% dosage, the stress at first crack increasedto 2.6 MPa. The MF40 fibers, on the other hand, had a lower first cracking stress of 2 MPa for 1% fiber dosage and an average of 3.8MPa and 4.4 MPa for the 2.5% and 4% replicates, as shown in Table 3. Regardless of the fibers used, stiffness, strength, and ductility increased significantly as the dosages increased. With increase in fiber content from 1%, 2.5%, and 4%; the pre-crack stiffness increased from $14.3 \pm 7.4$, to $21 \pm 3.3$, and $24 \pm 14$ GPa, respectively. The post-crack stiffness changes with the increase in the fiber content from $35 \pm 9$, $62 \pm 32$, to $177 \pm 28$ MPa for the MF fiber, which is much lower than the initial stiffness however much extends for a much larger strain range. The post-crack stiffnesses were significantly different in the two fiber systems, as shown by the reported values in Table 3. The crack spacing of composites at the crack saturation stage is shown in Figure 8c,d and point to the efficiency of the MF system.

The strain capacity of both macro- and micro-fiber systems, even at a 1% dosage level, exceeded 5%, which is an impressive level of deformation with significant energy absorption. The ultimate tensile strength for MAC composite replicates varied from 7.45 to 13.2 MPa. This tensile strength was significantly high and appropriate for the structural application of PP-based cementitious composites. Post-cracking stiffness increasedfrom 81 to 197 MPa over the entire strain range and depended on the various fiber contents. The overall toughness increasedfrom 0.79–0.83 MPa. MAC fibers at 1% and 2.5% fiber dosage showed similar stress–strain behavior, whereas at 4%, the overall composite stiffness and mechanical properties showed significant improvement. In all these systems, distributed cracking was the dominant mechanism, resulting in an increased overall toughness that was primarily due to a large strain range. The post-cracking behavior of the MF fibers were much improved compared to the MAC fibers, showing a stiffer post-crack response with distinct distributed cracking for 1% and 2.5% fiber dosages. However, for 4% MF40 dosage, several fine cracks close to each other resulted in a high crack density and toughness, as evident in Figure 8d. The first cracking stress for 2.5% and 4% fiber dosage of MF40 specimens were within 3.8 to 4.4 MPa;ultimate stress was 12.5 to 17.5 MPa; and toughness was 1.1 MPa to 1.37 MPa, which was 65% higher than MAC fibers at 4% dosage. The comprehensive result in these discussions can be found in Table 3.

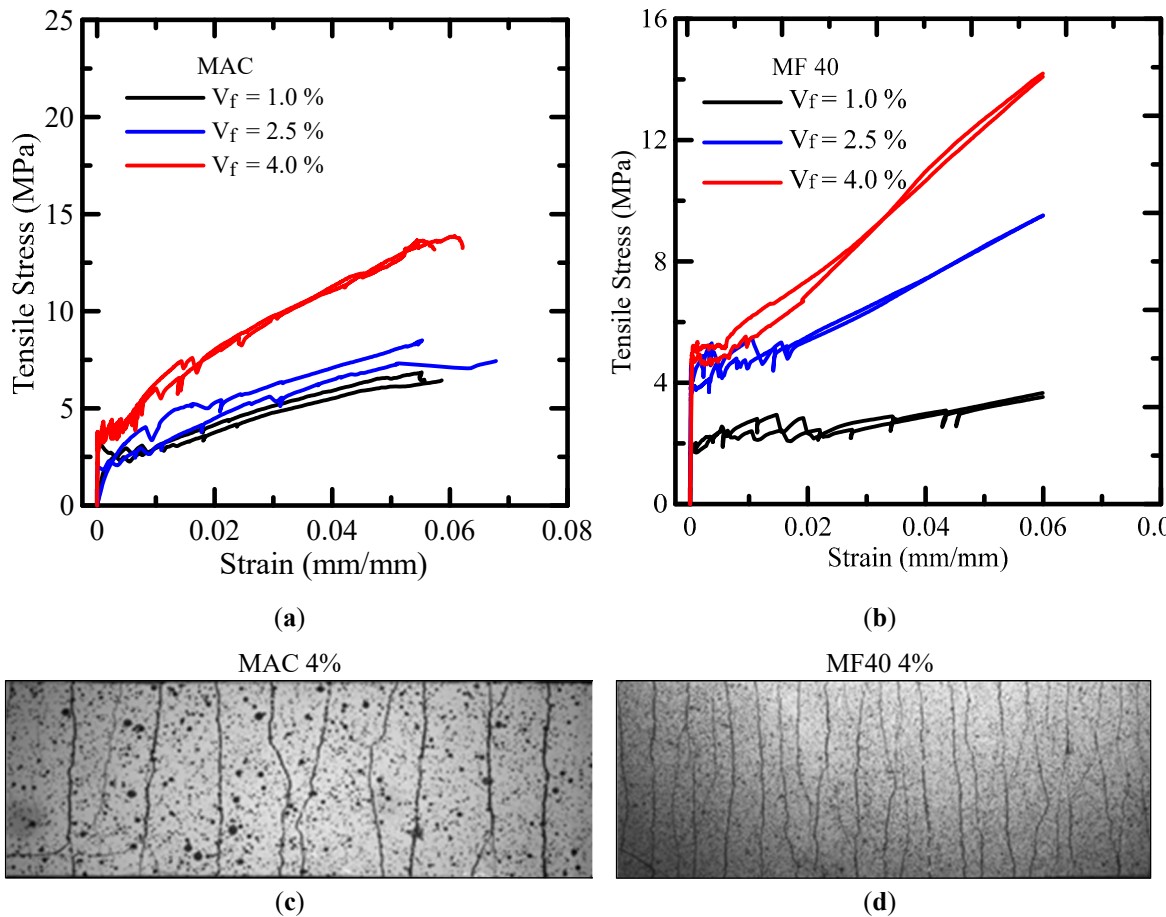

**Figure 8.** Effect of fiber content on the tensile stress–strain response: (**a**) MAC fiber and (**b**) MF40 fiber. Spacing distribution at crack saturation stage: (**c**) MAC composites and (**d**) MF40 composites.

Table 3. Stress Strain response parameters for control mix laminates with different fiber dosages.

| Fiber Type, Vf | Replicate ID | Stress at First Crack MPa | Strain at First Crack mm/mm | UTS MPa | Strain at UTS mm/mm | Ultimate Strain mm/mm | Young's Modulus (LVDT) GPa | Post-Crack Modulus MPa | Work to Fracture (Stroke) N·ms | Toughness at 5% Strain MPa | Toughness at 10% Strain MPa |
|---|---|---|---|---|---|---|---|---|---|---|---|
| | | | | | **MAC** | | | | | | |
| MAC 1% | #1 | 1.6 | $1.1 \times 10^{-3}$ | 7.4 | 0.12 | 0.16 | 1.4 | 94 | 90 | 0.16 | 0.43 |
| | #2 | 1.5 | $1.0 \times 10^{-3}$ | 7.3 | 0.13 | 0.16 | 1.5 | 70 | 89 | 0.14 | 0.41 |
| | #6 | 1.5 | $1.1 \times 10^{-4}$ | 8.3 | 0.13 | 0.15 | 13.6 | 80 | 65 | 0.09 | 0.37 |
| | Avg | 1.5 | $7.4 \times 10^{-4}$ | 7.4 | 0.13 | 0.16 | 2.8 | 81 | 89 | 0.15 | 0.42 |
| MAC 2.5% | #2c | 1.7 | $1.4 \times 10^{-3}$ | 8.6 | 0.13 | 0.17 | 1.2 | 88 | 143 | 0.16 | 0.48 |
| | #4 | 1.7 | $2.7 \times 10^{-4}$ | 5.0 | 0.07 | 0.12 | 6.4 | 110 | 57 | 0.10 | 0.32 |
| | #5 | 2.0 | $2.2 \times 10^{-4}$ | 8.8 | 0.12 | 0.16 | 9.1 | 180 | 131 | 0.17 | 0.51 |
| | Avg | 1.7 | $6.3 \times 10^{-4}$ | 7.5 | 0.11 | 0.15 | 7.6 | 126 | 110 | 0.15 | 0.44 |
| MAC 4% | #5 | 2.8 | $8.8 \times 10^{-5}$ | 12.7 | 0.08 | 0.09 | 32.0 | 170 | 146 | 0.31 | 0.78 |
| | #6 | 3.6 | $1.1 \times 10^{-4}$ | 12.9 | 0.08 | 0.11 | 33.6 | 170 | 205 | 0.35 | 0.89 |
| | #7 | 3.0 | $7.1 \times 10^{-5}$ | 14.1 | 0.10 | 0.10 | 42.2 | 250 | 160 | 0.31 | 0.79 |
| | Avg | 3.1 | $9.0 \times 10^{-5}$ | 13.2 | 0.08 | 0.10 | 16.9 | 197 | 170 | 0.32 | 0.82 |
| | | | | | **MF40** | | | | | | |
| MF40 1% | #2 | 2.1 | $1.8 \times 10^{-4}$ | 5.3 | 0.13 | 0.14 | 11.7 | 36 | 96 | 0.13 | 0.33 |
| | #3 | 2.7 | $1.6 \times 10^{-4}$ | 5.5 | 0.13 | 0.14 | 16.4 | 38 | 102 | 0.14 | 0.34 |
| | Avg | 2.4 | $1.7 \times 10^{-4}$ | 5.4 | 0.13 | 0.14 | 3.4 | 37 | 99 | 0.14 | 0.34 |
| MF40 2.5% | #3 | 3.1 | $1.3 \times 10^{-4}$ | 8.6 | 0.13 | 0.17 | 24.5 | 120 | 170 | 0.23 | 0.66 |
| | #5 | 4.5 | $1.1 \times 10^{-4}$ | 15.1 | 0.12 | 0.13 | 42.0 | 140 | 211 | 0.29 | 0.82 |
| | #8 | 5.2 | $1.2 \times 10^{-4}$ | 9.9 | 0.08 | 0.12 | 43.6 | 206 | 168 | 0.25 | 0.69 |
| | Avg | 4.3 | $1.2 \times 10^{-4}$ | 12.5 | 0.10 | 0.13 | 27.7 | 156 | 190 | 0.27 | 0.76 |
| MF40 4% | #3 | 4.7 | $1.5 \times 10^{-4}$ | 19.1 | 0.11 | 0.12 | 30.6 | 190 | 259 | 0.33 | 1.04 |
| | #5 | 5.1 | $1.2 \times 10^{-4}$ | 16.4 | 0.10 | 0.12 | 41.7 | 200 | 220 | 0.32 | 0.99 |
| | #7 | 5.2 | $1.1 \times 10^{-4}$ | 16.9 | 0.13 | 0.14 | 47.6 | 200 | 271 | 0.27 | 0.89 |
| | Avg | 5.0 | $1.3 \times 10^{-4}$ | 17.5 | 0.11 | 0.13 | 23.5 | 197 | 250 | 0.31 | 0.97 |

### 3.3. Digital Image Correlation

Digital image correlation (DIC) is a full-field displacement measuring approach that tracks the physical points of a speckle pattern on the surface of a specimen under deformation. Developed by Sutton et al. [20] and Bruck et al. [21], it is widely applied to experimental stress analysis [22–24]. For each subset region of a sample, the corresponding deformed position is found by searching in the vicinity that renders the correlation coefficient with the maximum likelihood or minimum cross-correlation function [23,24]. Commercial software VIC 3D-7 was used for measurement of crack density, spacing, and damage evolution [4,21,25,26].

Formation of a network of cracks and local strain fields are shown in Figures 9 and 10. The relative displacements of two points, as well as crack width and spacing parametersmeasured using the DIC, was compared with the LVDTs in Figure 9a,b. The DIC absolute and relative displacements along two horizontal segments were obtained at 10 s intervals and compared with the mean LVDT responses. The correlationwas close, as shown in Figure 9b, which validates the non-contacting DIC method since it provides a full-range response.

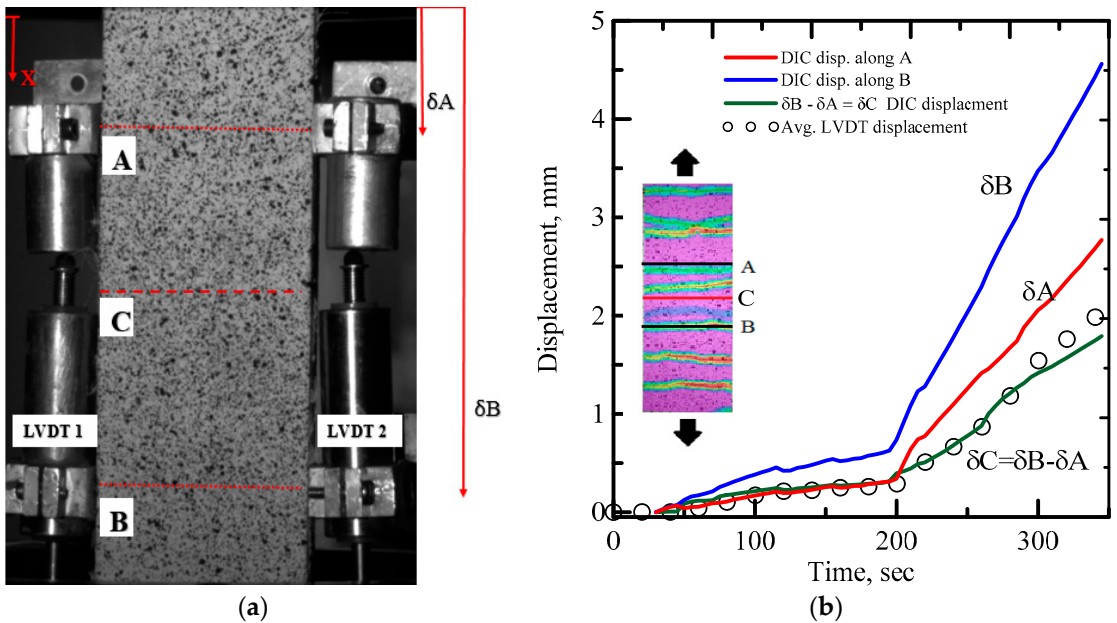

**Figure 9.** (**a**) Sample with LVDT mounted on sides. (**b**) DIC vs. LVDT displacement correlation.

The width of each crack was tracked from initiation to development to saturation stages and represented in Figure 10 showing the sequential formation of nine distributed cracks propagating throughout the width and observed as a function of time of a representative specimen. This data was post-processed to generate the crack width and spacing response up to the failure, as shown in Figure 10a,b representing the contour of longitudinal V displacement versus Y location for a MAC 4% (replicate 1) when all the cracks were formed. Each crack was marked as a discontinuity in displacement field, V(x), along with the Y location of the sample. The displacement discontinuity was measured as the crack width, as shown, and the crack spacing was marked as the distance along coordinate Y between any two cracks, as shown in Figure 10c.

Experimental stress versus time was compared with the crack formation, propagation, and widening, as shown in Figure 11a, using the individual crack openings measured using DIC post-processing. Results indicated that not all cracks were active at any given time during the loading history and the definition of strain may be significantly dependent on the gage length and the specific region of the specimen. Note that some of the cracks formed and then remained dormant before they opened further during subsequent loading stages.

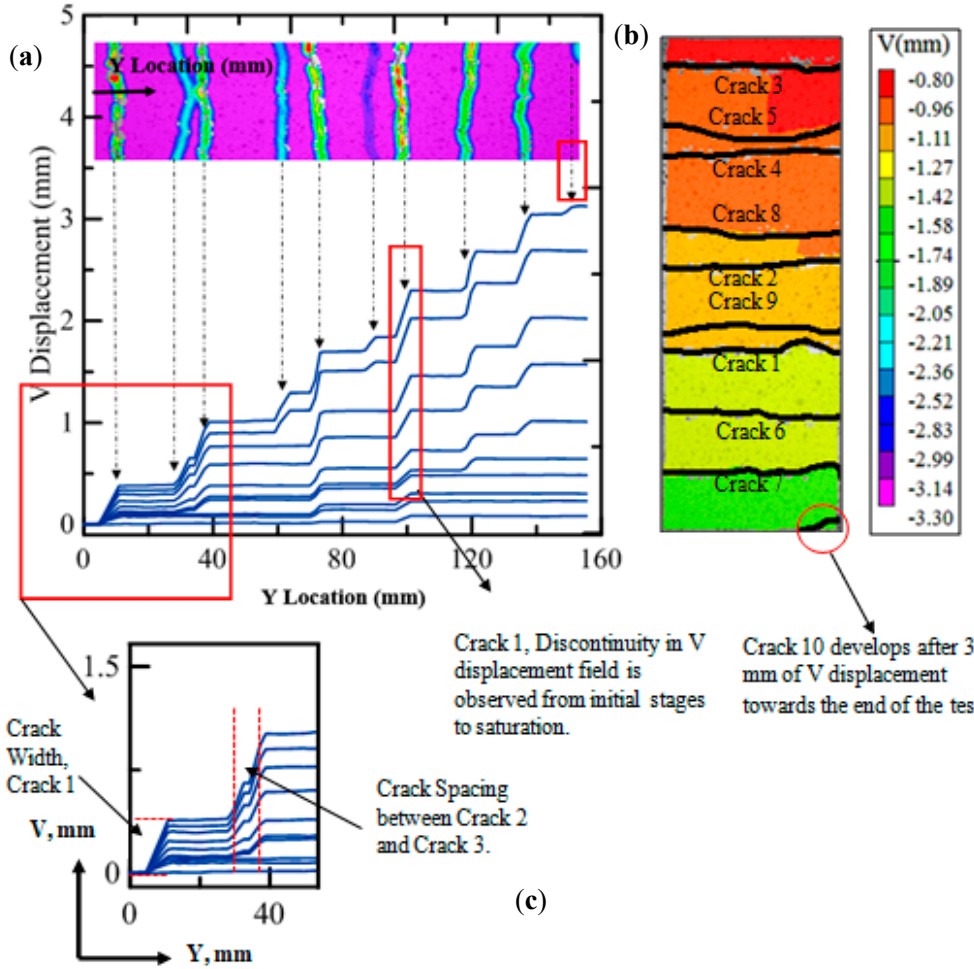

**Figure 10.** (**a**) Distribution of longitudinal strain is reported at distinct time steps, (**b**) DIC V displacement contour showing multiple crack formation at saturation stage, and (**c**) crack width and crack spacing estimation.

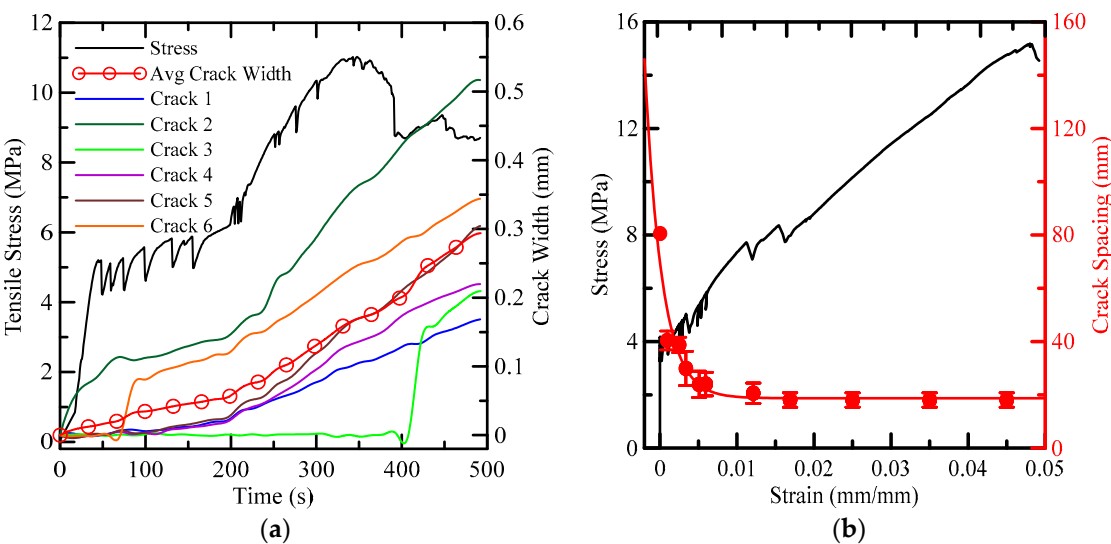

**Figure 11.** (**a**) Sequence of formation and individual stress–crack width response for MAC 4%, and (**b**) tensile stress–strain and crack-spacing response.

As shown in Figure 11a, Cracks 1, 5, and 3 developed early on within the first 100secs. of the test and widened within the time range of 250 to 400 secs. toreach a maximum 0.4 mm opening at the ultimate strength. However, Cracks 6 and 9 openedwhile the sample had reached its maximum stress capacity and the remaining cracks developed at a saturation crack width. The development of new cracks when the sample approachedmaximum stress indicates that multiple cracking stages of the overall composite was ending and the sample response was approaching the saturation stage. After the saturation stage, the majority of the cracks opened uniformly, indicating that the fiber phase was the primary load-carrying component. A stable crack spacing at this point and increased strains resulted in crack widening during the last stage offailure by fiber pullout. The crack spacing was measured as the distance between two cracks, as marked on the contour. At every strain, the number of cracks and their individual spacing was measured. The measure crack was plotted as function of applied strain and compared to tensile stress in Figure 11b. The mean crack spacing from these values indicatedthe damage induced at that point. An increasing strain reducedthe average crack spacing up to the saturation point. Figure 11b shows a saturation crack spacing of 20 mm at 0.015 strain. These data were further processed and shown as the relationship between crack spacing and applied strain.

### 3.4. Correlation of Fiber Size and Type on Crack Width and Spacing

The correlation of the representative stress–strain response with the distributed cracking on the two continuous fiber composites is shown in Figure 12. The filament structure of MF fibers developed the bond with the cementitious matrix. The interstitial spaces between the multiple filaments were used for penetration of the matrix phase, resulting in a superior mechanical bond and anchorage. The tensile strength exceeded 10 MPa, which was much higher than the mono-filament MAC fibers, which shows limited improvement in performance. A summary of the results of all the MAC and MF at different fiber contents are shown in Figure 13a,b, showing the correlation between the fiber content and crack saturation spacing measured from representative tests. With a tensile strength of about 8 MPa, the first cracking strength of MAC composites was quite similar to the plain matrix. The crack spacing–strain response implies that there were denser cracks with smaller individual crack spacing and lower saturated cracking MF fibers as compared to MAC fibers.

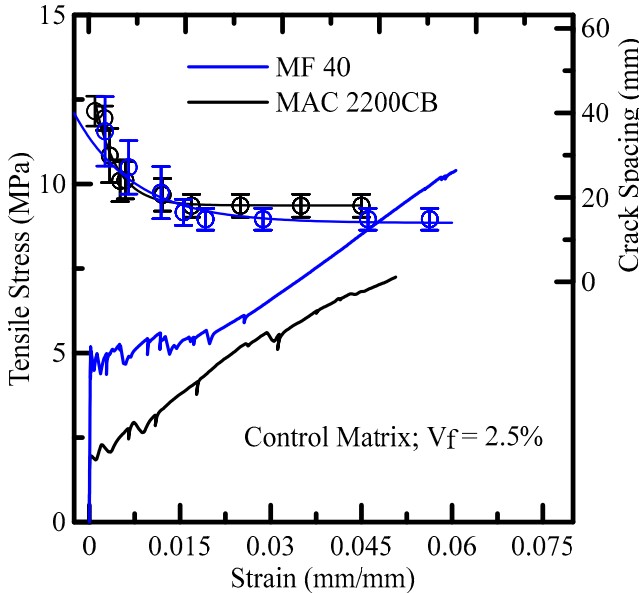

**Figure 12.** The tensile response of composites with MF40 fiber versus those with MAC fiber.

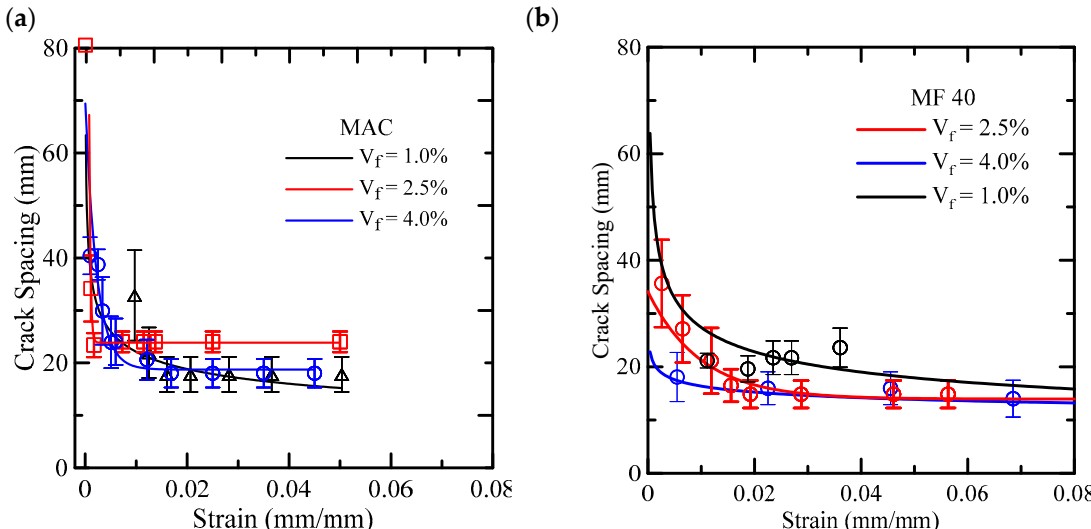

**Figure 13.** Tensile cracking behavior ofrepresentative fiber composites: (**a**,**b**) effect of tensile strain on the crack spacing formation for MAC and MF40 fiber composites, respectively.

The intensity of crack formation increased at higher dosages. The saturation crack spacing reduced from 25 to 15 mm with increasing fiber content, as shown in Figure 13a,b. At a dosage of 1%, the crack saturation was at 3% strain, while at dosages of 2.5% and 4%, new cracks continued to develop at higher strain levels of 6–7%, suggesting localized failure with lower dosages due to fewer cracks, and a dominance of crack-widening mechanisms.

### 3.5. Optical Microscopy

Toughening mechanisms were also observed by means of optical microscopy. The fiber reinforcement improved the ductility through several mechanisms that includedparallel cracking, crack bridging and deflection, fiber pullout, and fracture. The failure at the fiber–matrix interface was due to the transfer of shear stresses between the two phases, which exceeded the interfacial shear strength.

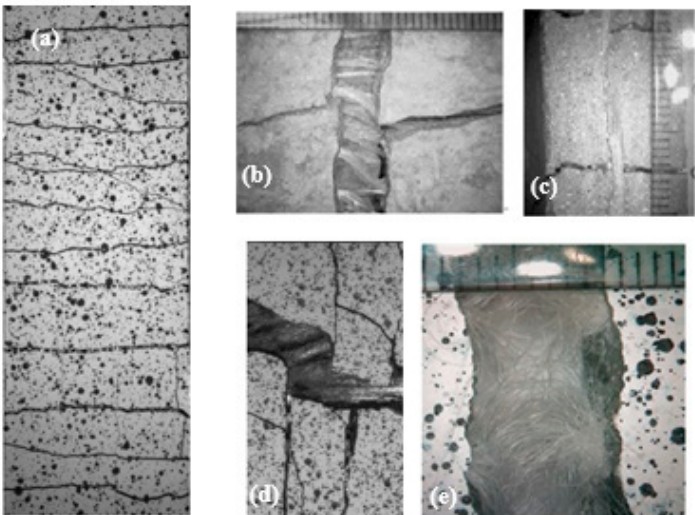

**Figure 14.** (**a**) Distributed saturation cracking in MAC 4% composites under tension, fiber bridging the crack (**b**) along the width and (**c**) along the thickness, (**d**) MF40 filaments debonding and pull out, and (**e**) MF40 filaments buckling after unloading (scale markers correspond to 1 mm).

The micrographs of multiple cracking are shown in Figure 14a–c for a representative MAC 4% specimen. Crack bridging is a key toughening mechanism within continuous fiber composites, which is shown along the width and thickness in Figure 14c. The fibers prevent significant strains and relaxation of the composite by bridging the distributed cracks and thereby slowing crack propagation, which allows for higher toughness. Figure 14d shows the delamination and pullout of MF40 fibers along with the transverse cracking with respect to the fiber direction. The superior bond exhibited by MF40 ledto fiber fractures accompanied by pullout. Figure 14e shows the crack bridging provided by MF40 fibrillated fibers, which began to buckle due to unloading as the matrix unloadedand compressedthe fibers. Fiber pullout was irreversible, and the final stage of tensile failurewas associated with unloading of the cracks and buckling of bridging fibers.

## 4. Conclusions

The effect of the macrosynthetic and bundled multifilament polypropylene fiber types on the distributed cracking and tensile stress–strain response of strain hardening cement composites were studied at different volume contents. Results indicate that tensile properties increased considerably with increasing fiber content. While the first cracking and ultimate tensile strength increased by about 200%, the post-crack modulus increased by over 400% as the volume fraction of microfilament micro MF fibers increased from 1–4%. Composites with monofilament macro MAC fibers with the increase in fiber content from 1–4% showed a more gradual increase of 100%, 78%, and 140% for first cracking, ultimate strength, and post-crack modulus, respectively. Comparing the two fiber types at 4% dosage, bundled microfibers exhibited higher first crack strength, ultimate strength, and toughness, which was 51%, 30%, and 65% higher than the mono-filament macro fiber systems. The tensile strength of the two systems compared at an average of 17.3 MPa versus 13.2 MPa for micro and macro fibers respectively. At the low fiber dosages, the performance of the macrofiber was slightly better. The nature of the open space between the multifilament structure of MF fibers allowed for penetration of the matrix and mechanical anchorage of the filaments, thus improving the interface bonding.

Four stages of composite stress–strain response consisting of the linear elastic stage upto the bend over point, the distributed cracking, and the crack widening zones were discussed in detail. The reduction of tensile stiffness during the distributed cracking provided for significant toughening and ductility of the composites. The general decrease in the crack spacing until saturation crack spacing was a key component of the material behavior. Evolution of crack spacing corresponding to the load was measured using quantitative DIC and correlated with the stress–strain response. At low dosages, crack formation was limited, and toughening through crack widening was more dominant. However at higher fiber dosages, especially with multifilament fiber yarns, denser crack distribution capacity and lower saturation crack spacing were observed. This enables better composite action with the cementitious matrix than the macrosynthetic fibers. The proposed structurally efficient, resilient, and durable sections promise to compete with several conventional building materials, such as timber and light gage steel, based sections for lightweight construction and panel applications.

**Author Contributions:** Conceptualization, B.M. and S.S.; Methodology, V.D. and J.B.; Software, V.D.; Validation, J.B. and H.M. Formal Analysis, B.M.; Investigation, V.D. and J.B.; Writing—Original Draft Preparation, VD.; Writing—Review & Editing, J.B.; Visualization, H.M.; Supervision, B.M. and S.S.; Project Administration, B.M.; Funding Acquisition, B.M.

**Funding:** This research was partially funded by the BASF corporation.

**Conflicts of Interest:** The authors declare no conflict of interest.The founding sponsors had no role in the design of the study; in the collection, analyses, or interpretation of data; in the writing of the manuscript, and in the decision to publish the results.

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
