# Peer review of "Reinforcing Efficiency of Micro and Macro Continuous Polypropylene Fibers in Cementitious Composites"

_applsci, doi:10.3390/app9112189_

Round 1
Reviewer 1 Report
The paper investigates the effects of microstructure of hydrophilic polypropylene fibers in distribution of cracking associated with strengthening and toughening mechanism of cement-based composites under tensile loading. The paper is clear and concise and it addresses an interesting and relevant problem. The English is excellent. The Figure and tables are well organized. The numerical approach developed in the manuscript is a good contribution to the field. In the light of that, I recommend in favour of publication of the paper in the journal. The paper might be accepted in the current form. However, I list some comments and suggestions that can be addressed by the authors while finalizing the manuscript in a minor revision process.
GENERAL COMMENTS:
1) The topic is well framed in the introduction section. I improve the discussion about DIC (which is adopted to address others relevant issue) improving the references with the following papers:
-) A numerical model based on ALE formulation to predict crack propagation in sandwich structures. Frattura Ed Integrita Strutturale, 13(47), 277-293. doi:10.3221/IGF-ESIS.47.21
-)Characterization of anisotropic polymeric foam under static and dynamic loading. Experimental Mechanics, 51(8), 1395-1403. doi:10.1007/s11340-011-9466-3.
2) I suggest to report the outline of the paper at the end of the introduction.
Author Response
1. The topic is well framed in the introduction section. I improve the discussion about DIC (which is adopted to address others relevant issue) improving the references with the following papers:
- A numerical model based on ALE formulation to predict crack propagation in sandwich structures. Frattura Ed IntegritaStrutturale, 13(47), 277-293. doi:10.3221/IGF-ESIS.47.21
- Characterization of anisotropic polymeric foam under static and dynamic loading. Experimental Mechanics, 51(8), 1395-1403. doi:10.1007/s11340-011-9466-3.
Response: These publications have been added
2. I suggest to report the outline of the paper at the end of the introduction.
Response: As reviewer has suggested, the outline of the paper has now been clearly presented at the end of the introduction.

Reviewer 2 Report
The work is interesting, some comments are as follows:
(1) In general, English should be improved.
(2) Regarding the title, “cementitious composites” could be better than “cement composites”.
(3) A typo in Line 37.
(4) Line 91, how did you obtain the effective yarn diameters and the surface to volume ratio?
(5) In Figure 2, some items indicated by the arrows are not clear.
(6) Line 130, give the reason why only the tension test results are discussed.
(7) Line 136, the unit of the strain rate should be s-1.
(8) Line 143, it is rare to use the expression of stress-strain properties, the stress-strain behavior is better.
(9) Line 145, what is the meaning of “yirding”?
(10) In Figure 4, for MF40, the curves show a turning point around 50 MPa, what is the reason behind?
(11) Provide a column more about the yield strength in Table 2; moreover, it looks like the results of max elongation are not reasonable considering the strain range in Figure 4. Furthermore, the unit of work to fracture is not appropriate.
(12) If compressive strengths of both materials are provided, the paper would be more valuable.
(13) In Figure 6, the points indicated by lines are not in the right position. Moreover, BOP is not appropriate, first crack strength is better.
(14) Adjust the width of the columns of Table 3. Moreover, the unit of toughness is not MPa.
(15) Line 257, where is “Table 6”?
(16) Provide some information more about DIC, such as resolution, size of subset, etc. Moreover, what is the frequency to take a photo?
(17) In Figure 10, some titles of axes are missing.
(18) In Figure 12, four sub-figures should be named individually.
Author Response
Reviewer #2:
3. In general, English should be improved.
Response: The paper has been carefully proofread, and the inconsistencies in the language have been corrected.
4. Regarding the title, “cementitious composites” could be better than “cement composites”.
Response: The word “cementitious” in lieu of “cement” has now been used in the title and several other sections in this paper.
5. A typo in Line 37.
Response: The typo “abnd” has been fixed to “and”.
6. Line 91, how did you obtain the effective yarn diameters and the surface to volume ratio?
Response: The effective yarn diameter was measured from image analysis of the SEM micrographs obtained during this study and verified with he manufacturer’s data.
7. In Figure 2, some items indicated by the arrows are not clear.
Response: The arrow in Figure 2 has now been fixed.
8. Line 130, give the reason why only the tension test results are discussed.
Response: In order to keep the length of the paper to a reasonable limit, only the tension tests were presented in this paper. We removed the reference to flexural tests from the section 2.2. The results from the flexural tests will be presented in a different paper.
9. Line 136, the unit of the strain rate should be s-1.
Response: The tension test on the fibers, was actually displacement controlled instead of strain controlled. The sentence has been modified.
10. Line 143, it is rare to use the expression of stress-strain properties, the stress-strain behavior is better.
Response: The sentence has been modified.
11. Line 145, what is the meaning of “yirding”?
Response: This was a typo. The word “yielding” has been used instead.
12. In Figure 4, for MF40, the curves show a turning point around 50 MPa, what is the reason behind?
Response: There was an issue during the post-processing of the raw data and not an actual feature from the test. That kink in the curves for MF40 fibers have been removed now through smoothening of data.
13. Provide a column more about the yield strength in Table 2; moreover, it looks like the results of max elongation are not reasonable considering the strain range in Figure 4. Furthermore, the unit of work to fracture is not appropriate.
Response: The fibers do not have a clear yield point, hence the ultimate tensile strength has been reported, which could be used for any design applications. Work to fracture measured from the area under the load-displacement curve from the actual tensile test of the fibers. Hence the unit of measure is N.mm, however as suggested more conventional energy unit of Joules has now been presented in Table 2 (1J = 10-3 N.mm).
We can see clearly that the strain at max stress from Figure 4(a) for MF 40 is in the range of 0.2 mm/mm, whereas for MAC it is only at about 0.1 mm/mm? This may be more valuable parameter. Your thoughts? Yes sure, they may be better than the strain alternatively you can call it Nominal maximum strain figure 4.b is there to show the effect of sequential failure of individual filaments and th easocated larfge displacement capacity of the specimen.
14. If compressive strengths of both materials are provided, the paper would be more valuable.
Compressive strength in individual yarns were not of interest due to the complexities involved in applying compressive stresses onto a fiber yarn. Due to the small diameter of the fiber, the failure is by means of buckling which is dominated by the end conditions and free length of the fibers.
15. In Figure 6, the points indicated by lines are not in the right position. Moreover, BOP is not appropriate, first crack strength is better.
Response: We agree with Reviewer’s observation and the Figure 6 was modified to callout First Crack Strength instead of BOP.
16. Adjust the width of the columns of Table 3. Moreover, the unit of toughness is not MPa.
Response: The width of the columns of this table has been adjusted to fit within the page. Toughness was measured as the area under the tensile stress-strain curve. Hence the unit of toughness measured is MPa.
17. Line 257, where is “Table 6”?
Response: Table 3 has been referenced in this sentence instead of Table 6.
18. Provide some information more about DIC, such as resolution, size of subset, etc. Moreover, what is the frequency to take a photo?
Response: 10 photographs were taken every second however the selected points corresponding to the specific strain levels were used.
19. In Figure 10, some titles of axes are missing.
Response: The Figures 10(a) and 10(b) were modified to show the axis titles clearly.
20. In Figure 12, four sub-figures should be named individually.
Response: In the title of Figure 12 was modified to correctly identify the context of the sub-figures.

Reviewer 3 Report
The paper is interesting and detailed. My comments are minor:
There are several typos, just in first page I found 2! Please check carefully.
Figure 9 a,b,c missing and legent has a format issue. Generally, in many figures (a), (b) etc are not visible
If possible add a reference regarding the fiber type (eg in producers web site)
In introduction discuss the possible application in real civil engineering project.
Polypropylene fibers are used against spalling in fire too. The bellow publication
can be useful
Maraveas, C.,Vrakas, A.A., Design of concrete tunnel linings for fire safety, Structural Engineering International, Vol. 24, Issue 3, pp 319-329, 2014. DOI: 10.2749/101686614X13830790993041
Please improve your conclusions. Include quantitative information regarding the evaluation and explanation of your test results, proposals for further investigation, research limitations and suggestions for engineering practice.
Author Response
Reviewer #3:
The paper is interesting and detailed. My comments are minor:
21. There are several typos, just in first page I found 2! Please check carefully.
Response: Please see our response to comment 3.
22. Figure 9 a,b,c missing and legent has a format issue. Generally, in many figures (a), (b) etc are not visible.
Response: The sub-figures in Figure 9 and other figures in this paper have been labeled correctly as (a), (b), etc.
23. If possible add a reference regarding the fiber type (eg in producers web site)
Response: Reference to the technical document on the MAC2200CB fiber was updated, see Reference #18. MF 40 is a recently developed fiber product by the manufacturer, and hence information available on it is limited currently.
24. In introduction discuss the possible application in real civil engineering project.
Response: A section on possible application of these cementitious composites have been in the introduction portion of this paper.
25. Polypropylene fibers are used against spalling in fire too. The bellow publication
can be useful
Maraveas, C.,Vrakas, A.A., Design of concrete tunnel linings for fire safety, Structural Engineering International, Vol. 24, Issue 3, pp 319-329, 2014. DOI: 10.2749/101686614X13830790993041
26. Please improve your conclusions. Include quantitative information regarding the evaluation and explanation of your test results, proposals for further investigation, research limitations and suggestions for engineering practice.
Response: Quantitative information regarding effect of fiber structure, and dosage on composite tension properties have been presented in the conclusion section. Brief write up on potential future testing and scope for this material in the real world engineering applications have also been presented at the end of this section.

Round 2
Reviewer 2 Report
My concerns were addressed completely in the revised manuscript.